# Investigating the Shear Strength of Granitic Gneiss Residual Soil Based on Response Surface Methodology

**DOI:** 10.3390/s23094308

**Published:** 2023-04-26

**Authors:** Hao Zou, Shu Zhang, Jinqi Zhao, Liuzhi Qin, Hao Cheng

**Affiliations:** 1The 3rd Geological Department of Hubei Provincial Geological Bureau, Huanggang 438000, China; 2Hubei Provincial Key Laboratory of Resources and Ecological Environment Geology, Wuhan 430034, China; 3Badong National Observation and Research Station of Geohazards, China University of Geosciences, Wuhan 430074, China; 4China Railway Siyuan Survey and Design Group Co., Ltd., Wuhan 430063, China

**Keywords:** granitic gneiss residual soil, shear strength, response surface methodology

## Abstract

The shear strength of granitic gneiss residual soil (GGRS) determines the stability of colluvial landslides in the Huanggang area, China. It depends on several parameters that represent its structure and state as well as their interactions, and therefore requires accurate assessment. For an effective evaluation of shear strength parameters of GGRS based on these factors and their interactions, three parameters, namely, moisture content, bulk density, and fractal dimension of grain size, were selected as influencing factors in this study based on a thorough investigation of the survey data and physical property tests of landslides in the study area. The individual effects and interaction of the factors were then incorporated by implementing a series of direct shear tests employing the response surface methodology (RSM) into the regression model of the shear parameters. The results indicate that the factors affecting shear parameters in the order of greater to lower are bulk density, moisture content, and fractal dimension, and their interactions are insignificant. The proposed model was validated by applying it to soil specimens from other landslide sites with the same parent bedrock, showing the validity of the strength regression model. This study demonstrates that RSM can be applied for parameter estimation of soils and provide reliable performance, and is also significant for conducting landslide investigation, evaluation, and regional risk assessment.

## 1. Introduction

Landslides are the major type of geohazards affecting the living environment of the residents in the Huanggang area, China. By the end of 2021, a total of 585 landslides have been recognized in the Huanggang area, which severely threaten human life and property safety [1]. The gneiss in the area has two main types, granitic gneiss and hornblende feldspar gneiss. In particular, landslides with bedrock of granitic gneiss account for 30%, and that of hornblende feldspar gneiss account for 25%, as shown in Figure 1. Upon field investigation, these colluvial landslides are predominantly minor or medium-sized, and primarily slide along the contact zone between weathered residual soil and bedrock [2]. Therefore, the shear strength of gneiss residual soils and its influencing factors are particularly essential for studying the stability of colluvial landslides in the Huanggang area.

Granite gneisses are formed by the recrystallization of minerals from granite under long-term ground stress, and their lithology and mineral composition are comparable to that of granite [3]. Being derived from the in-situ weathering and decomposition of granite gneisses, it is identified that the properties of granite gneisses residual soil (GGRS) are dominated by the parent rock [4], and hence the findings on the mechanical properties of granite residual soils (GRS) can be directly applied to the study of GGRS, thus the following text would employ the word residual soil to collectively describe these two kinds of soil. As a remark of commonality, the intrinsic microstructure, i.e., primary pores and fractures, as well as the unique grain distribution from clay to gravel, pose a significant challenge to sophisticatedly capturing the physical, hydraulic, and mechanical behaviors of residual soil [5,6,7,8,9].

The acquisition of the shear strength of the residual soil, of the mechanical properties, is of the most significance in the evaluation of colluvial landslide stability. The studies associated with the predominant factors governing the shear strength are thus of particular importance. Zhao et al. analyzed the strength of GRS by triaxial tests and direct shear tests, and found that the GRS have the characteristics of shear dilation and shrink, and concluded the effects of different particle compositions on shear strength [10]. Wu et al. identified the interaction effect of particle composition and matrix suction on the shear strength of GRS through laboratory experiments [11]. Wei et al. examined the shear strength properties of GRS based on the study of GRS in Southern China and indicated that the moisture content has a significant effect on their shear strength [12]. Meanwhile, a considerable number of studies on colluvial landslides in the Huanggang area also demonstrated that the reduced shear strength caused by the increasing moisture content of GGRS under heavy rainfall acts is the dominant factor for the instability [13]. Combined with related research results of other soil types, it revealed that the shear strength of GGRS is highly associated with its moisture content [12,14,15], compactness state [16], and particle composition [14,16,17,18]. However, the contribution of these factors and their interactions has been insufficiently studied.

To evaluate the correlation of shear strength of geo-materials and the related indices, the orthogonal experimental design method has been employed by the present studies. For instance, Zhou et al. studied the shear strength of soil-rock mixture under the freeze–thaw cycle environment considering five factors, including rock content, compaction degree, moisture content, number of freeze–thaw cycles and freezing temperature at four experimental levels, respectively according to the orthogonal experimental design method [19]. Ren et al. performed a series of multi-factor orthogonal softening experiments on gypsum rock from Lower Triassic Jialing River Formation [20]. However, the orthogonal test has the drawback of incapability of identifying the interaction of the different factors. Therefore, the interaction of the factors has not been identified. 

Given this shortcoming, the response surface methodology (RSM), a mathematical and statistical technique that optimizes the experimental results by approximately developing a explicit polynomial expression, provides a corresponding strategy that enables the complicated connection between the required response and input factors to be ascertained [21]. Moreover, it allows a minimization of the number of experiments and the level of independent variables, and offers internal error estimates [22]. It has been extensively applied in the food industry, chemical industry, etc., for optimization design [23]. Recently, a couple of studies have introduced it into the geotechnical engineering field for estimating the mechanical properties of geo-material and building materials. Asadizadeh et al. assessed the individual and interactive weighting contributions of the parameters (e.g., bridge length, bridge step angle, Joint Roughness Coefficient, etc.) on the shear and uniaxial compression strengths of jointed rocks through RSM [24]. Soltani et al. applied RSM to investigate the influence of cement content, water–cement ratio, and aggregate size on the compressive strength, permeability, and porosity of pavement concrete [25]. A recent study has also investigated the shear strength of paddy soil and its influencing factors by RSM method [17]. However, the application of RSM to the study of strength parameters of residual soils from colluvial landslides has not been reported yet.

There are two primary aims of this study: (1). To investigate the contribution of physical indices, in terms of moisture, dense state and grain size gradation, to the shear strength of GGRS; (2). To facilitate the empirical estimation model of shear strength of GGRS in a more effective way. In this sense, this study employed RSM to examine the individual and interacting effects of the factors. Moreover, the non-linear regression relationships were developed and verified to estimate the shear strength parameters via Analysis of Variance (ANOVA). The attempt of this application provides a prerequisite for the estimation of landslide stability in the study area, thus promoting the evaluation of landslide stability, and allows the extension of this research framework to other regions with similar geo-material.

## 2. Materials and Methods

### 2.1. Materials

The Huanggang area is located in the eastern Hubei Province, China. It has a border with the northern bank of the middle reaches of the Yangtze River and the southern foot of the Dabie Mountains. Generally, it has a high terrain in the northeast and a low terrain in the south. The region is a component of the Qinling stratum area’s eastern extension, which has relatively complete strata from the Archean to Cenozoic eras. With more than 1200 mm of annual precipitation, the average annual temperature is 16.3~18.2 °C. It is reported a population of around 7.49 million and an area of 17,453 km^2^ of the area [1].

#### 2.1.1. Overview of the Sampled Colluvial Landslides

Based on the comprehensive geotechnical survey data of the studied area, this study took GGRS specimens from five landslide sites for laboratory experiments. These five landslides are the Pingtouling landslide in Luotian County (PLL), the Guanshan welfare house landslide in Xishui County (GLX), the Qingcaoping landslide in Qichun County (QLQ), the Zoujiashan landslide in Macheng County (ZLM), and the Chengmagang kindergarten landslide in Huanggang town (CLH), as shown in Figure 1. The above landslides all have a lithology from top to bottom of GGRS and granitic gneiss. Among them, the GGRS specimens from PLL were selected for preparing specimens for response surface experiment design.

#### 2.1.2. Composition and Structure of GGRS

The composition and content of clay minerals in the GGRS play an overwhelming role in the stability of the colluvial landslides. X-ray diffraction analysis (XRD) was performed on the GGRS specimens, and the compositions and contents of the main mineral are presented in Table 1 and Figure 2. It is clear that the GGRS specimens in the Huanggang area contain a considerable amount of clay minerals, such as montmorillonite, rectorite, and illite, etc. These clay minerals are susceptible to swelling and softening with water, and thus are not conducive to slope stability [26].

Through the SU8010 field emission scanning electron microscope, the flake-like silicate minerals can be clearly observed in the form of void-filled intergranular voids, as shown in Figure 3. Under the action of fluid flushing, the adsorbed water and interlayer water inside the minerals enable the groundwater caused by the rainfall infiltration to be retained in the soil, increasing the soil weight; also, the swelling, dispersion, and cohesion produced by the action of minerals and water would reduce the mechanical strength of the soil, thereby reducing the stability of the colluvial landslides [27].

The particle gradation curves were examined by sieving and densitometer methods on the GGRS specimens (shown in Figure 4), and the percentage content of each particle group is shown in Table 2. The inhomogeneity coefficient, *C_u_*, and curvature coefficient, *C_c_*, of the soil samples were determined from the particle gradation curves, and it was indicated that *C_u_* > 5 and *C_c_* = 1~3 for all samples, indicating that the GGRS specimens are well graded.

#### 2.1.3. Parameters Considered for RSM Experimental Design

Previous studies have demonstrated that the water contents, dense state, and particle gradation of GGRS have significant impacts on shear strength. In this study, therefore, three indicators, water content, bulk density, and particle fraction dimension, were selected as the major physical indicators for consideration. Among them, the water content, ω, reflects the moisture of the soil in its natural state. The bulk density, *ρ_b_*, is the dry weight of the solid per unit volume of the solid, representing the density of the soil when there is no water in the pores at all. Similar to the porosity ratio, it reflects the degree of compactness of the soil. The particle fractal dimension, *D*, is the most significant index to quantify the complexity and irregularity of an object or fractal body, and is a parameter to quantitatively depict the degree of fractal self-similarity, which is generally defined by Equation (1).
(1)$$P_{i}={\left(\frac{d_{i}}{d_{\max}}\right)}^{3-D}\cdot 100$$
where, *P_i_* is the cumulative mass fraction of particles smaller than *d_i_*, *d*_max_ is the largest dimension of the particles. The *D* value herein was calculated and averaged according to the contents of gravel grains (>2 mm), sand grains (0.075~2 mm), silt grains (0.002~0.075 mm), and clay grains (<0.002 mm) presented in Table 2, according to Equation (2):(2)$$\mathrm{lg}\frac{M\left(<d\right)}{M_{0}}=\left(3-D\right)\mathrm{lg}\left(\frac{d}{d_{\max}}\right)$$
where, *M*_0_ is the total weight of each soil grain; *M*(<*d*) is the accumulated weight of soil with grain size less than *d*; *d*_max_ is the average diameter of the largest particle size; *d* is the grain size of soil. The larger the value of *D*, the larger the content of fine particles, and the lower the content of coarse particles in the soil.

The above three parameters of all the GGRS specimens from five landslide sites were counted, and the maximum, minimum, and mean value of the parameters are presented in Table 3.

### 2.2. Methods

#### 2.2.1. Direct Shear Tests

All the direct shear tests were performed on prepared GGRS specimens from studied landslides sites employing a tetragenous strain-controlled direct shear apparatus (TT-ADS4D, Figure 5a) in China University of Geosciences, Wuhan. The apparatus is equipped with a high precision stepper motor to apply vertical consolidation pressure and shear stress. It allows four simultaneous direct shear tests on soil samples with shear rates ranging from 0.001 to 2 mm/min, and automatically collects and transmits data to a computer system for results processing. The specimen with a diameter of 61.8 mm and a height of 20 mm was first consolidated at a normal pressure of 400 kPa for 24 h until stable consolidation, and then placed on the base of the shear box (Figure 5b), followed by fixing the upper and lower shear boxes, the pressurization system, and the measuring system to begin the test. Four GGRS specimens were applied with vertical pressures of 100 kPa, 200 kPa, 300 kPa, and 400 kPa, and sheared simultaneously at a rate of 0.08 mm/min, and the shearing of the specimens ceased when the shear displacement was up to 6 mm. The stress-displacement curves (Figure 5b) and the maximum shear stress under different normal stresses were then captured as the shear strength, and the shear strength components (*c* and *φ*) can be identified by linear fitting of determined shear strengths of GRSS specimens under different normal stresses according to the Mohr-Coulomb theory. The shear strength of soil is defined as: τ=c+tanϕ, where *c* is the shear strength with zero normal stress, and it depends on various physicochemical forces between soil particles, including Coulomb force, van der Waals force, cementation force, etc.; *φ* is the internal friction angle, and it relies on the occlusal friction and sliding friction between particles.

#### 2.2.2. RSM Method

The underlying idea of RSM is to formulate a polynomial to express the implicit functional by approaching. Essentially, it is a statistical approach with which to search for the best response value after considering the variability or uncertainty of the input independent variables. It determines the optimal conditions for a multivariable system rapidly and efficiently by designing a reasonable number of tests with few trials to precisely investigate the relationship between each independent factor and the response dependent value.

Commonly adopted design methods are Box-Behnken experimental design (BBD) and central composite design (CCD). The BBD method based on a spherical space design allows an efficient combination of variable factors with a minimum number of trials [28]. Compared to the CCD method, BBD is applicable to experiments with less factor levels, and the design is typically regarded as more efficient when dealing with three to five factors, avoiding the extreme value points that may lead to failure or instability of the test results.

In this study, the BBD method was adopted to design the influence factors. According to the results of the laboratory experiments as stated in Table 3, each influence factor had a low-level value (−1), a medium-level value (0), and a high-level value (1), as depicted in Table 4. 

The RSM approach, which can be regarded as a multivariable regression analysis, may be expressed as follows based on the factorial model of multiple quadratic regression equations.
(3)$$Y=\beta_{0}+\sum_{i=1}^{3}\beta_{i}X_{i}+\sum_{i=1}^{3}\beta_{ii}{X_{i}}^{2}+\sum_{i=1}^{3}\sum_{j=i+1}^{3}\beta_{ij}X_{i}X_{j}$$
where *Y* is the response shear strength index, represented by cohesion (*c*) and frictional angle (*φ*) in this study. *β*_0_, *β_i_*, *β_i_*_i_, and *β_ij_* are the regression coefficients, the first is a constant, and the following three denote the linear, quadratic and interactive coefficients, respectively. *X_i_* and *X_j_* are the independent influencing variables, represented by moisture content, bulk density, and fractional dimension in this study. *X_i_*^2^ and *X_ij_* denote the secondary and interactive effects of the independent variables.

It is noted that although two shear strength indices (*c* and *φ*) are investigated herein, they are both component indices obtained by linear fitting of uniquely determined shear strengths of GRSS specimens under different normal stresses controlled by the three independent variables. There may be some mathematical or physical correlation between the two indices, but currently no definitive agreement in the academic community. This study, therefore, excluded the interaction between the two shear strength indices and conducted univariate multivariable regression analysis based on RSM experimental design.

#### 2.2.3. Analysis of Variance (ANOVA)

The multivariable ANOVA is employed herein since the interaction between the two dependent shear strength indices is not taken into account. The amount that each independent parameter contributes to the dependent parameters may be determined using analysis of variance (ANOVA). The F-test is usually performed for this purpose to calculate the weighted contribution of each factor. The contribution of the variance from various sources to the total variance is analyzed to determine the magnitude of the influence of controllable factors on the study results, that is, the total variance is decomposed into the components of the individual variance, and then the significance test is utilized to make an appropriate judgement [24,29].

## 3. Results

### 3.1. Experimental Data

Based on BBD design method, a total of 13 sets of direct shear tests were launched. The variation in the responding shear strength parameters with three factors are presented in Table 5.

### 3.2. Modeling Shear Strength Parameters

Two responding models were developed to explore the interaction between the independent variables, namely, moisture content (ω), bulk density (*ρ_b_*), and fraction dimension (D), as well as their impact on the response shear strength parameters, involving cohesion (*c*) and fractional angle (*φ*), as presented in Equations (4) and (5).
(4)$$c=493.3725-0.38391\omega -75.01726\rho_{d}-335.606351D-0.21944\omega \rho_{d}+0.24074\omega D-7.61111\rho_{d}D-0.00157\omega^{2}+39.58333{\rho_{d}}^{2}+66.37037D^{2}$$
(5)$$\varphi =772.43111-0.94492\omega -94.77477\rho_{d}-534.01451D-0.34529\omega \rho_{d}+0.68405\omega D-34.39506\rho_{d}D-0.013189\omega^{2}+75.69213{\rho_{d}}^{2}+110.11934D^{2}$$

For verifying if the developed response surface regression model is reliable and stable, the ANOVA analysis was carried out herein, and the results are shown in Table 6 and Table 7. According to *F*-test, the larger the *F*-value, the greater the evidence that there is a difference between the group means. Moreover, *p*-value corresponding to the *F*-value, is also available for determining whether the difference between group means is statistically significant. If the *p*-value is less than α = 0.05, the null hypothesis of the ANOVA could be rejected and it suggests that the related parameter is significant and the data will be credible when the probability level is >95% [30]. Therefore, the responding models are significant and reliable due to the *p*-values of the two models. Meanwhile, the *F*-values of the two models are 6.62 and 7.07, respectively, indicating that the model is statistically significant and able to capture the relationship between dependent strength and each influencing factor. 

To better verify the accuracy of the model, predicted *R*^2^ is employed to reflect the quality of the model. The predicted shear strength parameters versus the actual values obtained from the direct shear tests are plotted in Figure 6. It depicts a great agreement of the datum, with the *R*^2^ of 0.9226 and 0.9271 for c and *φ* values, respectively. It provides the further proves for the accuracy of the response model. As depicted in Figure 6, all the c and *φ* data points fall into the 95% prediction band, 53.3% of the c data points are within the 95% confidence band and 46.6% of the *φ* data points are within the 95% confidence band, indicating that the regression equations have a significant prediction effect.

Employing the above model, the contribution of the three independent variables, three interactive coefficients, and three quadratic coefficients on the shear strength parameters were all taken into account. The impact of these coefficients can also be captured from Table 6 and Table 7. With regard to the responding model of *c* values, Table 6 indicates that the *p*-value for *ρ_b_* (0.0011) is the least and the *p*-value for *ω* (0.0403) is relatively large, indicating that they both have significant effects, and the influence of *ρ_d_* is much larger than that of *ω*. In contrast, the third variable, fractal dimension *D*, the interactive coefficients (*ω × ρ_b_*, *ω × D*, and *ρ_b_ × D*), and the quadratic coefficients (*ω*^2^, *ρ_b_*^2^, and *D*^2^) have *p*-values greater than 0.05, indicating that the impact is insignificant. With regard to the responding model of *φ* values, Table 7 indicates that the independent variables (*ρ_d_* and *ω*) and the quadratic coefficient of *ρ_b_*^2^ all have significant effects due to the low *p*-values, which is different from that of the *c* value. A closer inspection of the *p*-values shows that *ρ_b_* has the most pronounced influence, followed by *ω* and *ρ_b_*^2^. Besides, the rest of the coefficients have insignificant effects. 

### 3.3. The Effect of Independent Variable and Their Interaction on the Shear Strength

For visual comparison of the multivariate analysis, Figure 7, Figure 8 and Figure 9 depict the contour and response surface of cohesion plots obtained by Equation (4). The three-dimensional (3D) and two dimensional (2D) plots of the response surface are used to describe the interactive effect of the independent influence variables on the cohesion of GGRS. For each interactive scenario when two independent variables were varied, the other was kept constant at its intermediate value. In the 2D plots, the straight lines which were perpendicular to the contours were employed to determine which independent factor plays a dominant role. If the straight line is at an angle of 45 to the *X*-axis (*Y*-axis), it indicates that the interactive effect of the two independent variables is pronounced and the roles of them are almost equivalent. If the straight line is parallel or approximately parallel to the *X*-axis (*Y*-axis), it indicates that the role of the factor represented by the *X*-axis (*Y*-axis) plays a dominant role, and the interactive effect of the two factors is unimpressive.

Figure 7 depicts the effect of bulk density and moisture content on the cohesion of GGRS. Typically, the increasing of bulk density and decreasing of moisture content enhance the cohesion significantly. Lines AB and CD present two distinct effects of the variables for high and low bulk density, respectively. Line AB, with an angle of 12° to the *Y*-axis, indicates that the enhancement effect of bulk density is much more pronounced than that of the moisture content when it has a high bulk density (close or greater than 1.4 g∙cm^−3^), and the interaction of the two variables is insignificant. On the contrary, Line CD indicating the effect at low bulk density takes a large deflection and has an angle of 7.8° to the *X*-axis. It indicates that a low bulk density (1.20~1.45 g∙cm^−3^) may weaken its effect, and the effect of moisture content becomes dominant with negligible interactive impact of the two variables.

Figure 8 indicates that the cohesion is decreased with the increasing and decreasing of fractal dimension and an intermediate value of fractal dimension is related to the minimum cohesion with the same bulk density. In contrast, the increasing of bulk density enhanced the cohesion greatly. Notably, Line EF has an inclination of 33.2° to the x-axis representing the bulk density, indicating that the interaction of the two variables is significant, and the bulk density is the dominant factor herein.

A similar pattern to the effect of fractal dimension in Figure 8 can also be observed in Figure 9, whereby an intermediate value of fractal dimension is associated with the minimum cohesion with the same moisture content. Moreover, it also presents that increasing moisture content has a decreasing effect on cohesion considerably, regardless of the value of the fractal dimension. Line GH has an inclination of 4.2° to the *X*-axis, representing the moisture content, indicating that interaction of moisture content and fractal dimension is inconspicuous.

The 3D response surface and 2D plots of the quadratic model for the effect of moisture content, bulk density, and fractal dimension on the internal friction angle are presented in Figure 10, Figure 11 and Figure 12 based on Equation (5).

As indicated by Figure 10, the internal friction angle decreases with the decreasing bulk density and increasing moisture content. From the 2D plot, Lines IJ and KL show the distinct effect for high and low bulk density, respectively. As indicated by Line IJ, with an angle of 10.6° to the *Y*-axis, the enhancement effect of bulk density is dominant when it has a high value (close to or greater than 1.4 g∙cm^−3^), and the interactive effect is non-significant herein. Whereas a low bulk density (1.20~1.45 g∙cm^−3^) may weaken the effect of bulk density and the effect of moisture content becomes dominant since Line KL deflects greatly towards the *X*-axis, with an angle of 6.3° to the *X*-axis, indicating that the interactive effect of the two variables become significant.

Figure 11 shows that the internal friction angle is reduced as the fractal dimension increases and decreases, and that a midpoint in the fractal dimension is associated with the minimal value of the internal friction angle at the same bulk density. On the other hand, the internal friction angle was substantially improved by an increase in bulk density. The bulk density is the major factor in this situation, as shown by Line MN’s inclination of 31.5° to the *X*-axis. This also indicates that the two factors interact significantly.

Comparing with Figure 11, Figure 12 shows that the effect of fractal dimension shares a similar pattern; namely, that the increasing and deceasing of fractal dimension has a decreasing effect on the internal friction angle and an intermediate fractal dimension is associated with the minimum internal friction angle with the same moisture content. Moreover, the inclination of OP towards the *X*-axis is 4.5°, indicating that the increasing effects of moisture content are much greater than the effect of fractal dimension and the interactions are insignificant.

### 3.4. Validation of the Regression Model

To validate the applicability of the strength model based on RSM and ANOVA, physical parameter tests and direct shear tests were conducted for the remaining four landslides (GLX, QLQ, ZLM, and CLH) with in-situ GGRS specimens. Meanwhile, the geotechnical survey data of five other landslides, namely, the Mensijia landslide in Wenquan town (MLW), the Jieling Village landslide in Muzidian town (JLM), the Sanmahe Village landslide in Caohe town (SLC), the Mianyangfan Village landslide in Kuanghe town (MLK), and the Xintianpu landslide in Baliwan town (XLB), were collected as well. The influence factors and the shear strength parameters, as well as the predicted shear strength parameters of the GGRS from the aforementioned nine landslides, are presented in Table 8. The comparison of the actual experimental values and predicted values are shown in Figure 13. It is evident that the experimental and predicted values are highly compatible, with the *R*^2^ of 0.9471 and 0.9535, for cohesion and internal friction angle, respectively. As depicted in Figure 13, all the c and *φ* data points fall into the 95% prediction band, 77.8% of the *c* and *φ* data points fall into the 95% confidence band. These observations indicated that the proposed strength model is significantly effective and applicable for the GGRS distributed in the study area.

## 4. Discussion

The current study found that although there are slight differences in the degree of variation, the effect of the individual factors and their interactions on the cohesion and internal friction angle share a very similar pattern. Of these, bulk density has the greatest effect, followed by moisture content, while the effect of the fractal dimension of grain size has the least. One unanticipated finding was that the increasing moisture content has a considerable weakening effect on both cohesion and internal friction angle. This finding is contrary to previous studies which have suggested that the moisture only affects the cohesion, whereas it has an insignificant effect on the internal friction angle [31]. A possible explanation for this might be stated as follows. Owing to the larger moisture content, the orientation of fine particles during the shearing process is more pronounced, and leads to the weakening of interparticle bonding, and the reduction in larger pores between the agglomerates and the growth of smaller pores. Combined with the lubricating effect of free water, it leads to the reduction in both cohesion and internal friction angle as two inseparable components of the shear strength [32]. Moreover, the discrepancy could be attributed to the different fine grain content of the studied fine samples. The high content of fines results in more significant bond weakening effect and lubrication effect by free water, which in turn affects the interparticle friction and manifests in the reduction in the internal friction angle component.

Meanwhile, contrary to expectations from the ANOVA that there is no significant interaction of the variables, the further analysis of the responses surface indicated that the interaction between bulk density and fractal dimension is more considerable, and bulk density has a certain interaction with moisture content, while the interactive action between moisture content and fractal dimension is negligible. This might be related to the definition of bulk density, which corresponds to the density of a soil when it is completely waterless in its pores. Its conversion relationship with other variables is [33]
(6)$$\rho_{b}=\frac{\rho }{1+\omega }=\frac{G_{s}}{1+e}$$
where *ρ* is the natural density; *G_s_* is the specific gravity, which is associated with the mineral composition and particle gradation; and *e* is the porosity. 

It is noticeable that bulk density is naturally dependent on the moisture content and particle gradation (fractal dimension in the current study), and therefore coincides with the results of the above analysis. Thereby, it is further demonstrated that relying on ANOVA solely fails to yield robust findings, and the response surface model successfully captures this interactive effect, even if it is not significant in terms of manifesting in *p*-values. This finding, while preliminary, suggests that the interaction of the key factors is necessary, and the proposed response surface model provides an effective solution for shear strength estimation.

Future studies on the current topic are therefore recommended. Firstly, as noted above, this study regarded the shear strength indices as unrelated dependent variables and performed univariate multivariable regression analysis. The multivariate regression analysis may be facilitated in the further study considering the covariance matrix analysis among the dependent variables; and the other multivariate analysis, such as principal component analysis, cluster analysis, and MANOVA may be adopted to capture more fresh results [34]. Secondly, due to the drawbacks of the direct shear test, the further study may be implemented by the triaxial compression or ring shear apparatus, with emphasis on observing the change in stress-strain during the test, as well as stiffness and residual shear strength. Moreover, the shear strength of the geomaterials in the study area can be estimated based on the estimation model so that stability analysis can be launched to analyze the spatial-temporal landslide hazard [35,36].

## 5. Conclusions

This study performed a series of laboratory experiments to investigate the main contribution of individual physical variables, moisture content, bulk density, and fractal dimension of grain size, and their interaction to the shear strength parameters of GGRS of colluvial landslides in the Huanggang area, China. The conclusions can be drawn as follows.

Firstly, employing RSM approach and ANOVA analysis, a prediction model for the shear strength parameters of GGRS is proposed based on the factorial model of multiple quadratic regression equations considering the individual and interacting effects of the considered three variables, moisture content, bulk density, and fractal dimension of grain size. With regard to either cohesion or internal friction angle, bulk density has the greatest effect, followed by moisture content, while the effect of the fractal dimension of grain size is the least. Combined with the response surface plots, other than the interaction of bulk density and fractal dimension of grain size, the interaction of other variables is insignificant.

Secondly, the prediction model for the GGRS is validated by performing laboratory experiments to the GGRS sampled from other colluvial landslides with the same parent rock, as well as the collected data. The model is proved to be significant and applicable to the same kind of GGRS of the area.

This statistical analysis explored the variables at three separate levels and assess the efficacy of each parameter and the interactive coefficients comprehensively. The further study may extend the experimental design method to CCD to explore the possibility of extreme value. Moreover, the statistical method can be applied to more influencing variables and more objective properties, e.g., the permeability and compression index, to obtain more sophisticated observations. Besides, the proposed prediction model for GGRS can be applied to assess the regional landslide stability considering the infiltration process of the rainfall on the surface.

## Figures and Tables

**Figure 1 sensors-23-04308-f001:**
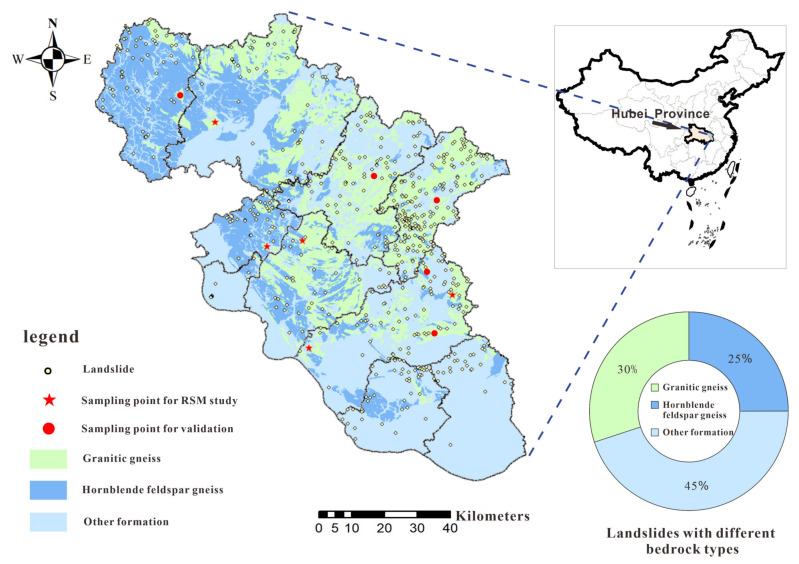
Landslides distributed in different lithological formations in the Huanggang area.

**Figure 2 sensors-23-04308-f002:**
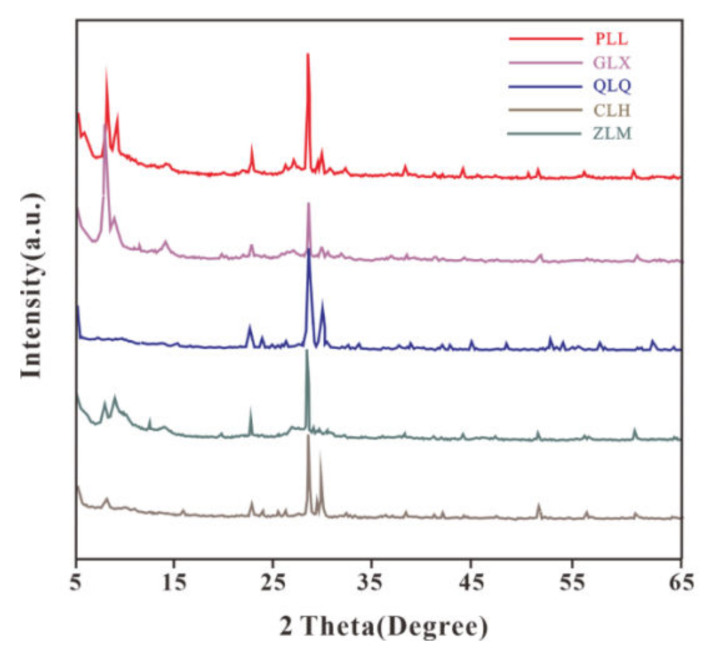
Mineral composition and content of the GGRS specimens.

**Figure 3 sensors-23-04308-f003:**
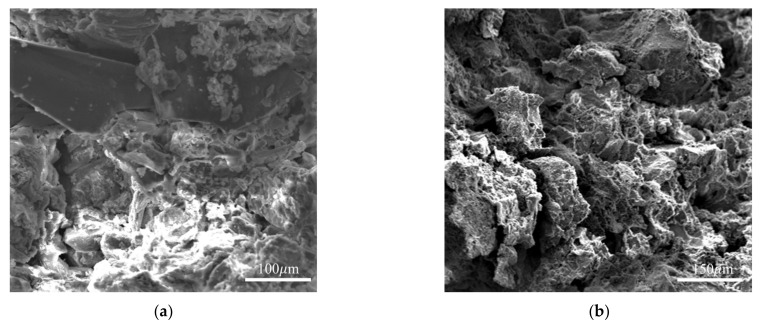
SEM images of GGRS specimens: (**a**) flake-like silicate minerals in specimens obtained from PLL; (**b**) intergranular voids observed in specimens obtained from PLL.

**Figure 4 sensors-23-04308-f004:**
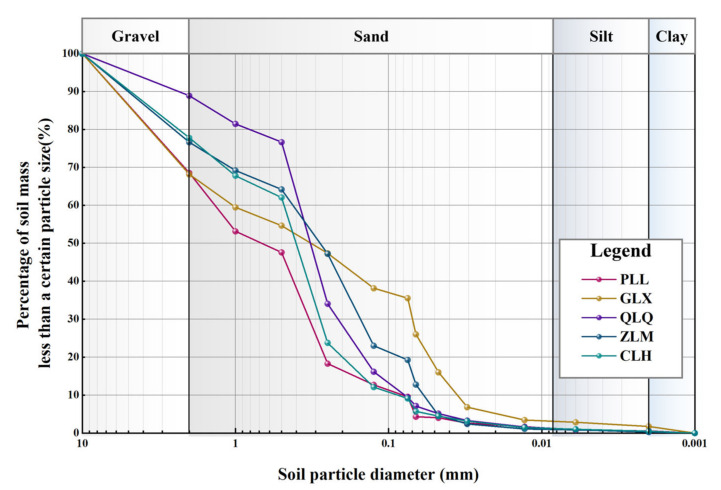
Particle gradation curves of GGRS specimens.

**Figure 5 sensors-23-04308-f005:**
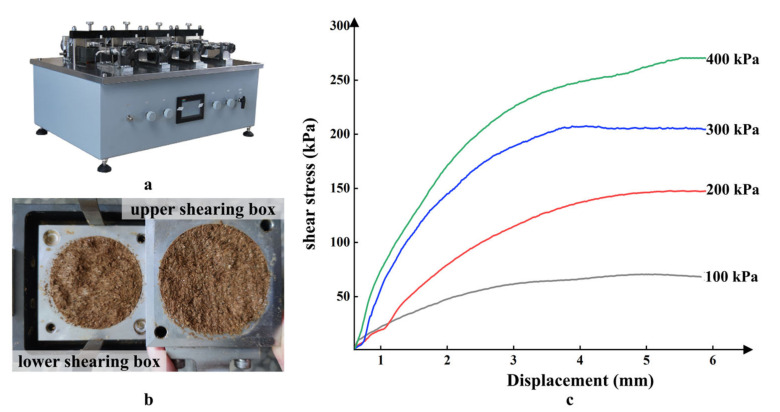
Direct shear test: (**a**) tetragenous strain-controlled direct shear apparatus (TT-ADS4D); (**b**) GGRS specimen in the shearing box after shear test; (**c**) representative stress-displacement curves, the colored lines and the adjacent pressure values represents the stress-displacement curves at diverse normal stress.

**Figure 6 sensors-23-04308-f006:**
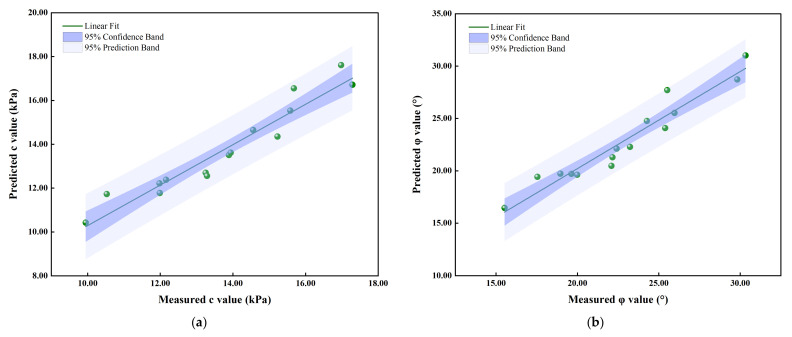
Predicted shear strength parameters versus actual values obtained from direct shear tests: (**a**) predicted *c* values versus measured *c* values; (**b**) predicted *φ* values versus measured *φ* values.

**Figure 7 sensors-23-04308-f007:**
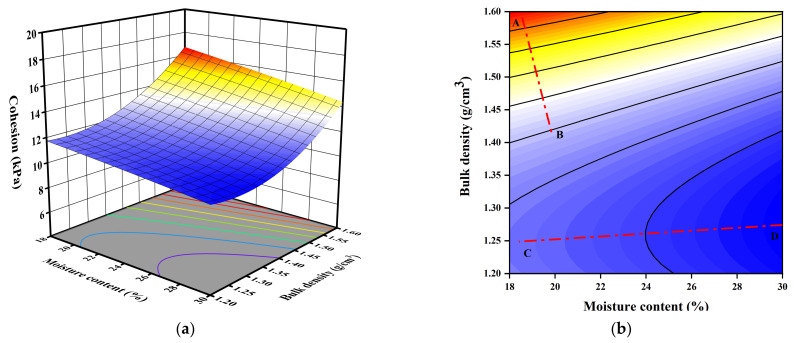
3D and 2D response surface plots of the interactive effect of moisture content and bulk density on cohesion: (**a**) 3D plot; (**b**) 2D plot.

**Figure 8 sensors-23-04308-f008:**
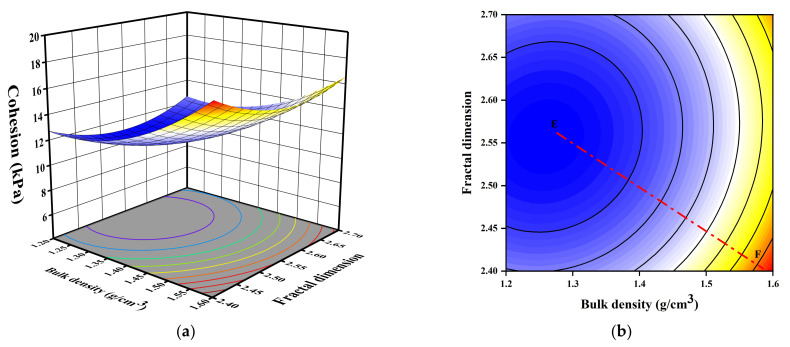
3D and 2D response surface plots of the interactive effect of bulk density and fractal dimension on cohesion: (**a**) 3D plot; (**b**) 2D plot.

**Figure 9 sensors-23-04308-f009:**
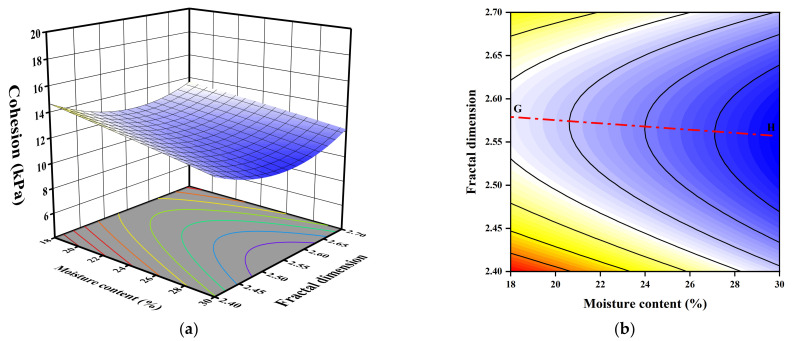
3D and 2D response surface plots of the interactive effect of moisture content and fractal dimension on cohesion: (**a**) 3D plot; (**b**) 2D plot.

**Figure 10 sensors-23-04308-f010:**
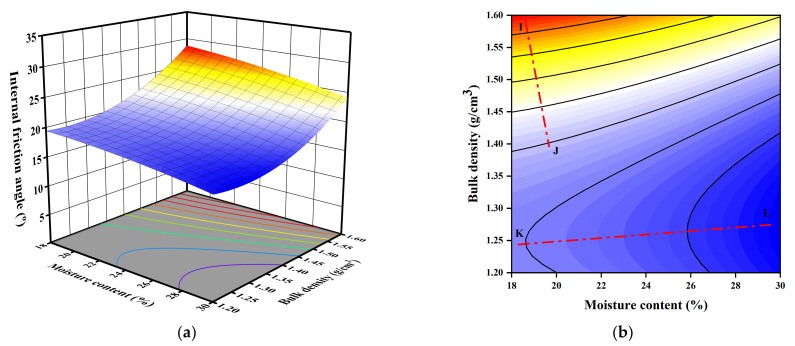
3D and 2D response surface plots of the interactive effect of bulk density and moisture content on internal friction angle: (**a**) 3D plot; (**b**) 2D plot.

**Figure 11 sensors-23-04308-f011:**
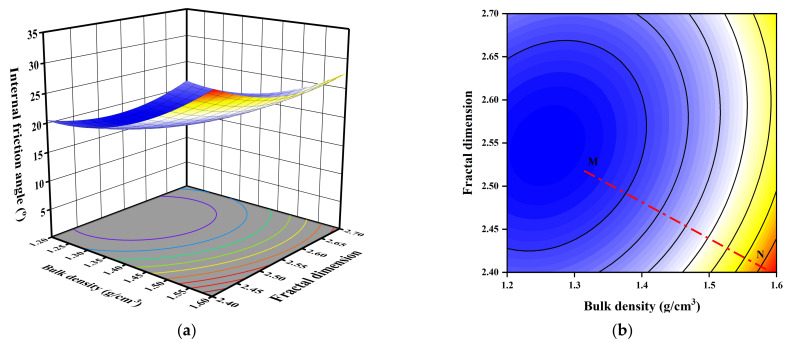
3D and 2D response surface plots of the interactive effect of bulk density and fractal dimension on internal friction angle: (**a**) 3D plot; (**b**) 2D plot.

**Figure 12 sensors-23-04308-f012:**
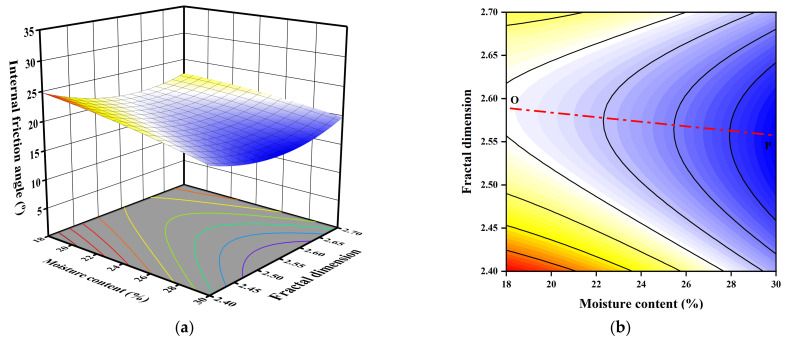
3D and 2D response surface plots of the interactive effect of moisture content and fractal dimension on internal friction angle: (**a**) 3D plot; (**b**) 2D plot.

**Figure 13 sensors-23-04308-f013:**
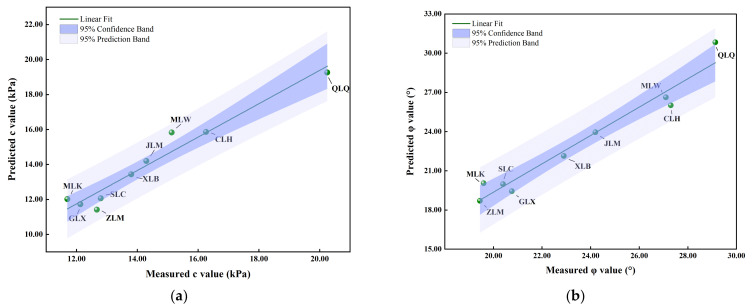
Predicted shear strength parameters versus actual values obtained from direct shear tests and collected data: (**a**) predicted *c* values versus actual *c* values; (**b**) predicted *φ* values versus actual *φ* values.

**Table 1 sensors-23-04308-t001:** Mineral composition of the GGRS specimens.

Sample No.	Percentage of Mineral Composition (%)						
	Montmorillonite	Rectorite	Illite	Tremolite	Quartz	Orthoclase	Albite
PLL	37.74	23.54	—	—	21.12	5.75	11.85
GLX	65.36	13.9	—	3.1	10.64	—	7.01
QLQ	3.34	—	27.36	—	33.11	—	36.2
ZLM	17.92	33.46	—	12.87	28.54	—	7.22
CLH	13.64	—	16.15	—	31.79	10.66	27.76

**Table 2 sensors-23-04308-t002:** Percentage content of each grain group.

Sample No.	Percentage Content of Each Grain Group (%)				*C_u_*	*C_c_*
	>2 mm	0.075–2 mm	0.002–0.075 mm	<0.002 mm		
PLL	31.49	58.97	8.99	0.55	21.822	1.146
GLX	31.83	32.65	33.73	1.79	19.818	0.077
QLQ	11.08	79.50	9.05	0.37	4.951	1.521
ZLM	23.37	57.34	19.01	0.28	8.036	0.877
CLH	22.21	68.66	8.64	0.49	6.868	2.594

**Table 3 sensors-23-04308-t003:** Statistics of physical parameters of the samples.

Parameters	*ω*/%	*ρ_b_*/g · cm^−3^	*D*
Maximum	26.28	1.53	2.76
Minimum	23.27	1.20	2.37
Mean value	24.92	1.37	2.55

**Table 4 sensors-23-04308-t004:** Influence factors levels of RSM experimental design.

Levels	*ω*/%	*ρ*/g · cm^−3^	*D*
−1	18	1.2	2.4
0	24	1.4	2.55
1	30	1.6	2.7

**Table 5 sensors-23-04308-t005:** BBD design layout and corresponding response of shear strength of GGRS.

Run	*ω*/%	*ρ*/g · cm^−3^	*D*	Response	
				c/kPa	*φ*/°
1	24	1.4	2.55	11.99	19.99
2	18	1.2	2.55	10.53	17.55
3	18	1.6	2.55	17.29	29.82
4	30	1.4	2.7	11.98	19.63
5	24	1.6	2.4	16.98	30.33
6	24	1.2	2.7	13.29	22.16
7	18	1.4	2.7	13.94	23.23
8	24	1.2	2.4	13.25	22.09
9	30	1.6	2.55	15.23	25.39
10	30	1.2	2.55	9.95	15.53
11	24	1.6	2.7	15.68	25.52
12	18	1.4	2.4	14.56	24.27
13	30	1.4	2.4	12.16	18.96

**Table 6 sensors-23-04308-t006:** ANOVA analysis for cohesion response surface regression model.

Source	Sum of Squares	Degree of Freedom	Mean Squares	*F*-Value	*p*-Value	Performance
Model	61.79	9	6.87	6.62	0.0255	significant
A-*w*	7.84	1	7.84	7.57	0.0403	significant
B-*p_b_*	46.34	1	46.34	44.71	0.0011	significant
C-*D*	0.84	1	0.84	0.81	0.4103	insignificant
AB	0.33	1	0.33	0.31	0.5993	insignificant
AC	0.22	1	0.22	0.21	0.6640	insignificant
BC	0.24	1	0.24	0.24	0.6475	insignificant
A^2^	8.459 × 10^−3^	1	8.459 × 10^−3^	8.161 × 10^−3^	0.9315	insignificant
B^2^	6.60	1	6.60	6.37	0.0529	insignificant
C^2^	5.87	1	5.87	5.67	0.0631	insignificant
Residual Error	5.18	5	1.04			
Total	66.97	14				

**Table 7 sensors-23-04308-t007:** ANOVA analysis for friction angle response surface regression model.

Source	Sum of Squares	Degree of Freedom	Mean Squares	*F*-Value	*p*-Value	Performance
Model	228.63	9	25.40	7.07	0.0222	significant
A-*w*	33.78	1	33.78	9.40	0.0279	significant
B-*p_b_*	167.30	1	167.30	46.53	0.0010	significant
C-*D*	3.60	1	3.60	1.00	0.3628	insignificant
AB	0.81	1	0.81	0.22	0.6558	insignificant
AC	1.78	1	1.78	0.50	0.5131	insignificant
BC	5.00	1	5.00	1.39	0.2914	insignificant
A^2^	0.59	1	0.59	0.17	0.7012	insignificant
B^2^	24.15	1	24.15	6.72	0.0487	significant
C^2^	16.17	1	16.17	4.50	0.0874	insignificant
Residual Error	17.98	5	3.60			
Total	246.61	14				

**Table 8 sensors-23-04308-t008:** Experimental results of predicted and experimental values.

Test Group	*w*	*ρ_b_*	*D*	Measured		Predicted	
				*c*	*φ*	*c*	*φ*
GLX	24.7	1.322	2.66	12.13	20.76	11.72	19.43
QLQ	13.1	1.617	2.74	21.25	29.13	19.26	30.83
ZLM	30	1.157	2.64	13.67	19.45	11.41	18.70
CLH	20.1	1.5	2.75	16.26	32.32	15.86	26.01
MLW	35.7	1.047	2.77	15.13	27.17	15.82	26.62
JLM	18.8	1.505	2.57	14.32	24.26	14.19	23.94
SLC	27.4	1.463	2.54	12.81	20.44	12.06	19.97
MLK	21.8	1.363	2.49	11.73	19.69	12.01	20.04
XLB	26.1	1.456	2.70	13.84	22.98	13.43	22.14

## Data Availability

Not applicable.

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
