# Peer review of "Investigating the Shear Strength of Granitic Gneiss Residual Soil Based on Response Surface Methodology"

_sensors, 2023, doi:10.3390/s23094308_

Round 1

Reviewer 1 Report

The analytical part of the manuscript is well presented and detailed. Besides, two points must be modified.

1) the description of the method (meaning the experiment and the instrument used: 2.2.1 )is poor. Moreover, the figure 5 supporting the description is not complete and not easy to read and interpret, including the very short and not clear caption.

2) the multivariate approach, meaning a multivariate technique to evaluate the concurrent influence of all the varaiables and the variances, and not just a multidimensional representation of more than one parameter, is missed. 

Regarding figures axes are not readable

Reviewer 2 Report

Please clearly formulate the aim of the work.

Reviewer 3 Report

  • This manuscript presents a fresh investigation of properties of granitic gneiss residual soils based on response surface methodology. The proposed regression model is significant for landslide investigation, evaluation and regional risk assessment. I Think this manuscript should be accepted after a minor revise.
